# Identification of a Multi-Messenger RNA Signature as Type 2 Diabetes Mellitus Candidate Genes Involved in Crosstalk between Inflammation and Insulin Resistance

**DOI:** 10.3390/biom12091230

**Published:** 2022-09-02

**Authors:** Hebatalla Said Ali, Mariam Sameh Boshra, Sara H. A. Agwa, Mohamed S. Abdel Hakeem, Mahmoud Shawky El Meteini, Marwa Matboli

**Affiliations:** 1Medical Biochemistry and Molecular Biology Department, Faculty of Medicine, Ain Shams University, Abbassia, Cairo P.O. Box 11381, Egypt; 2Clinical Pathology, Medical Ain Shams Research Institute, Ain Shams University, Abbassia, Cairo P.O. Box 11381, Egypt; 3Institute of Immunology, University of Pennsylvania, Philadelphia, PA 19014, USA or; 4Department of General Surgery, The School of Medicine, University of Ain Shams, Cairo 11591, Egypt

**Keywords:** Type 2 Diabetes Mellitus, mRNA, bioinformatics, STING, NLR blood

## Abstract

Type 2 Diabetes Mellitus (T2DM) is a metabolic disease associated with inflammation widening the scope of immune-metabolism, linking the inflammation to insulin resistance and beta cell dysfunction. New potential and prognostic biomarkers are urgently required to identify individuals at high risk of β-cell dysfunction and pre-DM. The DNA-sensing stimulator of interferon genes (STING) is an important component of innate immune signaling that governs inflammation-mediated T2DM. NOD-like receptor (NLR) reduces STING-dependent innate immune activation in response to cyclic di-GMP and DNA viruses by impeding STING-TBK1 interaction. We proposed exploring novel blood-based mRNA signatures that are selective for components related to inflammatory, immune, and metabolic stress which may reveal the landscape of T2DM progression for diagnosing or treating patients in the pre-DM state. In this study, we used microarray data set to identify a group of differentially expressed mRNAs related to the cGAS/STING, NODlike receptor pathways (NLR) and T2DM. Then, we comparatively analyzed six mRNAs expression levels in healthy individuals, prediabetes (pre-DM) and T2DM patients by real-time PCR. The expressions of ZBP1, DDX58, NFKB1 and CHUK were significantly higher in the pre-DM group compared to either healthy control or T2DM patients. The expression of ZBP1 and NFKB1 mRNA could discriminate between good versus poor glycemic control groups. HSPA1B mRNA showed a significant difference in its expression regarding the insulin resistance. Linear regression analysis revealed that LDLc, HSPA1B and NFKB1 were significant variables for the prediction of pre-DM from the healthy control. Our study shed light on a new finding that addresses the role of ZBP1 and HSPA1B in the early prediction and progression of T2DM.

## 1. Introduction

Type 2 Diabetes Mellitus (T2DM) is a metabolic disease associated with inflammation widening the scope of immune-metabolism linking the inflammation to insulin resistance and beta cell dysfunction. There is a worldwide increase in the prevalence of T2DM with its complications including diabetic retinopathy, nephropathy, neuropathy and cardiovascular strokes, which are the main causes of morbidity and mortality related to T2 DM [1]. The prevalence of DM in Egypt in 2019 was almost around 9 million adult cases, occupying second place in the Middle East and North Africa (MENA) region [2,3].

T2DM diagnostic criteria that were clarified by the American Diabetes Association (ADA) include the following: A fasting plasma glucose (FPG) level of 126 mg/dL or higher, or A 2-h plasma glucose level of 200 mg/dL or higher, or A hemoglobin A1c (HbA1c) level of 6.5 or higher [4]. Although there are some difficulties in distinguishing type 1 and type 2 DM in all age groups at onset, the actual diagnosis becomes more obvious over time using autoimmunity-specific tests [5]. The problem is that most of the patients with type 2 DM do not have specific symptoms in the early stage and, once diagnosed, the majority of the cases will have serious complications. Prediabetic-state current laboratory methods show several limitations in the early prediction of pre-diabetes and T2DM. The diagnosis of pre-DM relies on the oral glucose tolerance test as a gold standard, but this test is time-consuming and complicated. Although fasting blood glucose is a convenient tool for T2DM diagnosis, the rate of missing pre-DM diagnosis is relatively high. In addition, HbA1c% is likely to be linked to other changes in red blood cell life rather than glycation rates, e.g., haemoglobinopathies. That is why there is an urgent need to find future potential biomarkers for pre-DM early detection [6]. During prediabetes (pre-DM), beta islets undergo stress and release many mRNAs [7]. Several etiological factors have been reported to have a role in the development of T2DM; one of them is the activation of the immune response as a result of overnutrition, leading to low-grade chronic inflammation, which may be a strong contributor to the development of T2DM [8].

The disbiosis of gut microbiota can lead to inflammatory changes with subsequent activation of nuclear factor-kappa B (NF-κB), which may be triggered by activation of cGAS/STING [9] or nucleotide-binding oligomerization domain-containing protein (NOD) receptor signaling, resulting in decreased insulin sensitivity and beta cell function [10].

STING (The DNA-sensing stimulator of interferon genes) is an important mediator of interferons inflammatory response, which can sense any foreign pathogen and activates protective antimicrobial signals [11]. The cGAS–STING–IRF3 pathway plays a role in metabolic stress-induced endothelial inflammation in obesity [12]. STING1 (TMEM173) also can recognize self-DNA leaking from the nucleus to the cytoplasm. Recently, evidence showed that STING plays an important role in many metabolic pathways as insulin resistance due to obesity, fat metabolism in the liver leading to Non-Alcoholic Fatty Liver [13,14]. STING is a critical regulator for both glucose and lipid metabolism. STING knockout significantly improved insulin resistance and glucose intolerance in rat on a high-fat diet [8]. Z-DNA-binding protein 1 (ZBP1) is a positive mediator of innate immunity through its cooperation with the cGAS-STING pathway [15]. ZBP1 is a cytoplasmic sensor of DNA and has an important role in the immune response activated by the introduction of different viruses inside the body [16]. ZBP1 can lead to the production of type 1 interferon through the activation of the interferon regulatory factor (IRF) and nuclear factor-kappa B (NF-kappaB) transcription factors [17]. Thus, it can promote chronic inflammation in various pathological conditions as insulin resistance [18]. DDX58 is an innate immune receptor that can detect cytoplasmic DNA and activate the signaling pathway, leading to the production of interferon 1 and different inflammatory cytokines [19]. A crosslink was found between cGAS-STING1- and DDX58-MAVS-dependent innate immune response pathways.

NOD is an intracellular pattern recognition receptor that recognizes fragments of the bacterial cell wall; when it is mutated, it loses the ability to respond properly to bacterial cell wall fragments with dysregulation in NF-κB signaling. Acute stimulation of NOD signaling by mimetics of bacterial PGNs causes insulin resistance [20]; NOD1 ligands lipid-derived metabolites were produced during obesity and contribute to insulin resistance development [21]. There is strong crosstalk between NOD signaling and the insulin receptor pathway through NF-KB and MAPK intermediates [22]. Among NOD stimulators, HSP70 can aggravate the response of NOD2 to bacterial cell wall fragments, and increase NOD-like receptor (NLR)-mediated NF-KB activation [23]. Heat shock proteins (HSPs) are proteins responsible for cellular stress response that inhibit denaturation or protein unfolding in response to stress. HSPs are linked to the modulation of several pathways in antigen-presenting cells like tumor antigen uptake and processing through MHC Class I and class II pathways. HSP60 showed a direct link between innate immunity and pancreatic islets functions [24]. Heat shock protein is related to many inflammatory diseases such as diabetes mellitus, rheumatoid arthritis and atherosclerosis [25]. Another crucial player of NOD signaling is conserved helix–loop–helix ubiquitous kinase (*CHUK*) encodes for Inhibitor-κB kinase α (IKKα) that acts as a catalytic domain of the Serine/Threonine kinase, Inhibitor of Nuclear Factor-KappaB Kinase (IKK). IKKα phosphorylate IκBα and inhibitors of NF-KB, leading to their degradation, thus activating NF-KB [26].

A literature search shed light on the crucial role of the STING and NLR pathways in development of inflammation-mediated insulin resistance. The DNA-sensing cGAS-cGAMP-STING pathway mediates type I interferon inflammatory responses in immune cells during infections. Recent studies showed that this pathway is also activated by host DNA aberrantly localized in the cytosol, contributing to increased sterile inflammation; insulin resistance via potential interactions of the cGAS-cGAMP-STING pathway with mTORC1 signaling and apoptosis have been discussed, suggesting its critical role in obesity-induced metabolic diseases [27]. STING trafficking and degradation are also regulated by a variety of mechanisms; for example, Nod-like receptors NLR family domain-containing protein 3 (NLRC3) interacts with STING to modulate its trafficking [28]. Moreover, obesity causes significant changes in the skeletal muscles and adipose tissue with an increase in the plasma free fatty acids (FFA). Also, obesity induces changing in gut microbiota composition. Both FFA and lipopolysachrides (LPS) trigger the activation of the Nod-like receptors (NLRs)-mediated inflammation, which further activates NF-kB with subsequent development of inflammation-mediated insulin resistance [29].

We propose exploring novel blood-based mRNA signatures that are selective for components related to inflammatory, immune and metabolic stress, which may reveal the landscape of T2DM progression for diagnosing or treating patients in the pre-DM state. In this study, we used a microarray data set to identify a group of differentially expressed mRNAs related to the cGAS/STING, NOD-like receptor pathways (NLR) and T2DM. Then, we comparatively analyzed six mRNAs expression levels in healthy individuals, prediabetes (pre-DM) and T2DM patients. We also demonstrated that these mRNAs may represent a multi-messenger RNA signature that can be used to effectively discriminate between healthy and pre-DM individuals to identify T2DM susceptible ones.

## 2. Materials and Methods

### 2.1. Selection of mRNA Set Linked to cGAS/STING, NOD-like Receptor Pathways (NLR) and T2DM

We have used several microarray databases including KEGG: Kyoto Encyclopedia of Genes and Genomes (https://www.genome.jp/kegg (accessed on 22 May 2022))), Human protein atlas (https://www.proteinatlas.org/ (accessed on 22 May 2022)) and GeneCards Human Genes database (https://www.genecards.org/ (accessed on 22 May 2022)). We have retrieved a set of mRNAs based on the following criteria: (a) Genes that are expressed and deregulated in T2DM; (b) Genes that are highly expressed in tissues of interest in T2DM, e.g.,: skeletal muscle, adipose tissue and also expressed in Peripheral Blood Mononuclear Cells (PBMCs) obtained from whole-blood samples for easiest extraction and least invasiveness; (c) These genes are related to the cGAS-cGAMP-STING pathway and NOD-like receptor signaling pathway, which are parts of the PAMPs and DAMPs pathways of innate immunity and chronic sterile inflammation; (d) Linking these novel genes from the 2 concerned pathways to previously known genes in T2DM through an interaction network. We have selected 6 mRNAs, namely: ZBP1, Heat Shock Protein Family A (Hsp70) Member 1B (HSPA1B), Stimulator of Interferon Response CGAMP Interactor 1(TMEM173), DExD/H-Box Helicase 58 (DDX58), Nuclear Factor Kappa B Subunit 1(NFKB1) and CHUK (Appendix A). The six selected 6 mRNAs were then imported into the Search Tool for the Retrieval of Interacting Genes (STRING; version 11.0; Zurich, Switzerland http://stringdb.org (accessed on 22 May 2022)) online database for protein–protein interaction (PPI) assessment (Appendix A).

### 2.2. Study Population and Blood Samples

This study was authorized by the medical research ethical committee, Ain Shams University, Faculty of Medicine. All subjects included in the study provided informed consent according to the Declaration of Helsinki. The study is formed of 44 individuals with a pre-DM group (pre-DM), 61 T2DM patients and 45 healthy control individuals, age- and sex-matched. The participants were selected from the Endocrinology Unit, Department of Internal Medicine, Faculty of Medicine, Ain Shams University during the period between February 2020 and March 2021. Patients were diagnosed according to the American Diabetes Association (ADA) criteria 2021 for the diagnosis of T2DM [30]. All patients with liver or kidney dysfunction, inflammatory diseases, cancer, autoimmune diseases and any endocrine disease other than T2DM were excluded from our study.

All the participants gave a detailed past history of any chronic medical conditions and full clinical examinations were done. Any participant with blood pressure above 139/89 mmHg was considered a hypertensive patient [31]. Anthropometric measures were obtained, Body Mass Index (BMI) was calculated according to the WHO recommendations [32]. After 8–10 h of overnight fasting, 10 mL of venous blood were taken from every participant and divided into three samples: 4 mL of them were used for quantitative colorimetric determination of glycated hemoglobin as a percent of total hemoglobin using kits supplied by Sigma-Aldrich, St. Louis, MO, USA [33]. Then, another 1 mL of blood was transferred into a sodium flouride tube, and another sample of blood was taken after 2 h for determination of blood glucose by enzymatic colorimetric method using Thermofischer, USA. The other 5 mL were transferred into a plain tube and samples were left to coagulate, then centrifuged at 1300× *g* for 20 min. The sera samples were separated and kept at −80 °C until biochemical analysis and RNA extraction. 

Biochemical analyses were carried out, including: fasting blood glucose levels, glycated hemoglobin (HbA1c%), lipid profile and Albumin/creatine ratio. HOMA-IR formula was calculated to assess the degree of tissue resistance to insulin. (Serum fasting insulin (SFI) in μmol/dL × Serum fasting glucose (SFG) in mmol/L) over 22.5. Insulin resistance was determined when the score was above 2.5 [34]. HOMA-B was used to assess insulin sensitivity. Serum total cholesterol (TC), Triglycerides (TG), HDL-c and LDL-c were estimated using a multifunctional biochemistry analyzer (AU680, Beckman Coulter Inc., Indianapolis, IN, USA). 

### 2.3. Extraction of the mRNA

The total RNA was extracted from the serum samples using the miRNeasy Serum/Plasma Kit extraction kits (Cat No. 217184, Qiagen, Hilden, Germany), according to the manufacturer’s protocol. The concentration and integrity of RNA in the different samples were measured using a Nano-Drop instrument (Thermo Scientific, Waltham, MA, USA), using samples with RNA: protein higher than 1.8–2.

RNA samples were stored at −80 °C till analysis. Then, reverse transcription was performed using miScript II RT Kit (Qiagen, Hilden, Germany, Cat No. 218161) to obtain cDNA following the instructions of the manufacturer. Four microliter 5 x miScript HiFlex Buffer, two microliter 10 x miScript Nucleics Mix, one microliter miScript Reverse Transcriptase Mix and RNase free water were added to 2 ug RNA extract, then incubated at 37 °C for 60 min, then 95 °C for 5 min using a Rotor gene Thermal cycler (Thermo Electron Waltham, MA, USA).

### 2.4. Quantitation of the Selected Six-Based mRNA Signature Expression

The differential expression of ZBP1, HSPA1B, TMEM173, DDX58, NFKB1 and CHUK, the 6 chosen mRNAs, was determined using Quantitect SYBR Green Master Mix (Qiagen, Hilden, Germany) and specific primers for (Hs_ZBP1_1_SG QuantiTect Primer Assay) (NM_00116041), (Hs_HSPA1B_1_SG QuantiTect Primer Assay) (NM_005346), (Hs_TMEM173_1_SG QuantiTect Primer Assay) (NM_198282), (Hs_DDX58_1_SG QuantiTect Primer Assay) (NM_014314), (Hs_NFKB1_1_SG QuantiTect Primer Assay) (NM_001165412), and (Hs_CHUK_1_SG QuantiTect Primer Assay) (NM_001278); following the manufacturer’s protocol, alongside with GAPDH (NM_002046) as the reference gene. The PCR program was as follows: initial activation step at 95 °C for 15 min followed by 40 cycles of PCR were done under the following conditions: at 94 °C for 15 s, at 55 °C for 30 s and at 72 °C for 30 s.

All the reactions were carried out in duplicate using The Rotor Gene real-time PCR detection system (Qiagen, Hilden, Germany). The amplification plot curve and melting curve were used to assess the specificities of the amplicons. If the threshold cycle exceeded 36, it was considered negative. Melting curves were used to identify the specificity of the amplicons of the PCR. The relative expression of the mRNAs was estimated by 2^−ΔΔCT^ [35]. The results of the samples were compared to a control sample and reference gene. 

### 2.5. Statistics 

The Statistical Package for the Social Sciences 20th version (SPSS, Chicago, IL, USA) was used to analyze all the statistical data. The Chi-square test, one-way ANOVA, and Kruskal–Wallis test were performed for comparing the samples. The cutoff of the mRNA as a predictor of T2DM was obtained using the receiver operating characteristic (ROC curves). The correlation between mRNA levels and the different clinicopathological variables were assessed using Spearman correlations. A 2-tailed *p* value ≤ 0.05 is considered statistically significant.

## 3. Results

### 3.1. The Study Groups’ Medical and Anthropometric Characters

The medical and anthropometric characters of all subjects are shown in Appendix A. We found that the FSG levels and the HbA1c are significantly higher in the group with T2DM compared to the healthy controls group and the group with impaired glucose level (pre-DM). As for the blood pressure, serum cholesterol and LDL-c were significantly higher in the T2DM group compared to the other 2 groups, and also higher in the group with pre-DM in relation to the healthy controls. On the other hand, the BMI was significantly higher in both the T2DM group and the pre-DM group in comparison with the healthy controls, as shown in Appendix A.

### 3.2. Insulin Sensitivity and the Function of the Pancreatic Islet Cells

The HOMA-IR was calculated to determine the presence of insulin resistance; it was significantly higher in the T2DM group in relation to the other 2 groups. Also, HOMA-IR was significantly higher in the pre-DM group in comparison with the healthy controls. Concerning the HOMA-B values, we found that it was significantly reduced in the T2DM group compared to the group with the pre-DM group and the healthy controls, it was also reduced in the group with pre-DM compared to its value in the healthy controls, as shown in Appendix A.

### 3.3. The Expression of ZBP1, HSPA1B, TMEM173, DDX58, NFKB1 and CHUK in the Sera Samples

The expression of ZBP1, HSPA1B, TMEM173, DDX58, NFKB1 and CHUK was estimated in the sera of all the study groups (Table 1). A Mann–Whitney test was performed; it was found that the 6 mRNAs were significantly highly expressed in the group with T2DM compared to the pre-DM group and the healthy controls. They were also significantly higher in the pre-DM group in relation to the healthy controls, as shown in Figure 1A–F. Of note, there was a significant progressive increase in the expression levels of ZBP1, HSPA1B, TMEM173, DDX58, NFKB1 and CHUK from healthy to pre-diabetic by 5, 3.5-, 40-, 84-, 14.3- and 31-folds, respectively), and from the pre-DM group; T2DM individuals by 10.5-, 3.7-, 19.5-, 9.5-, 23- and 3.6-folds, respectively).

Patients with T2DM were further divided into 2 groups regarding their glycemic control. HbA1c < 7% was considered good glycemic control, while HbA1c ≥ 7% was considered poor glycemic control. There was significant upregulation in the expression of (Zbp1 and DDX58) in the poor glycemic control group compared to individuals with good glycemic control (*p* = 0.0241 and 0.054, respectively). Interestingly, HSPA1B showed higher expression in patients with HOMA-IR ≥ 2.5 compared to the other group (*p* = 0.05) (Table 2).

We examined the degree of insulin resistance that could affect the expression of the mRNA panel, so, we divided the group with T2DM into two subgroups, one with insulin resistance (HOMA-IR ≥ 2.5) and the other subgroup being insulin sensitive (HOMA-IR < 2.5). HSPA1B mRNA showed a significant difference in its expression regarding the insulin resistance (Table 3).

The optimum cutoff values of ZBP1, HSPA1B, TMEM173, DDX58, NFKB1 and CHUK used for discriminating patients with T2DM and the control groups were calculated using ROC curves. ZBP1 was 3.25, HSPA1B was 4.7817, TMEM173 was 4.005, DDX58 was 7.750, NFKB1 was 6.525 and CHUK was 3.640 with sensitivity of 88.5%, 91.8%, 93.4%, 93.4%, 96.7%, 96.7%, respectively, as shown in Figure 2A–F.

ROC curve analysis was performed to discriminate between the groups with pre-DM and T2DM; the optimum cutoff values of ZBP1, HSPA1B, TMEM173, DDX58, NFKB1 and CHUK were 15.590, 6.898, 12.751, 26.5700, 17.350 and 17.740 with sensitivity of 75.4%, 85.2%, 90.2%, 85.2%, 90.2%, 73.8%, respectively, as shown in Figure 3A–F.

ROC curve analysis has been also used to discriminate between the healthy control group and the group with pre-DM. The optimum cutoff values of ZBP1, HSPA1B, TMEM173, DDX58, NFKB1 and CHUK were 1.600, 2.341, 1.800, 1.220, 1.580 and 2.050 with sensitivity of 88.6%, 81.8%, 95.5%, 97.7%, 88.6%, 93.2%, respectively, as shown in Figure 4A–F.

The best combination was that of ZBP1, TMEM173 and NFKB1 to discriminate between pre-DM and the development of T2DM, with sensitivity of 100%, specificity of 68.2% and accuracy of 86.6%. Linear regression analysis for prediction of pre-DM revealed that the most significant variable for prediction of preDM was LDLc (*p* value 0.001), (Standardized Coefficients Beta 0.429 followed by HSPA1B (*p* value 0.014), (Standardized Coefficients Beta 0.118); then NFKB1 (*p* value 0.031), (Standardized Coefficients Beta 0.113) (Appendix A).

### 3.4. Correlation Analysis

There was high significant positive correlation between the 6 chosen mRNAs among all studied groups and also among the T2DM group and pre-DM group, as seen in Table 4. Thus, we can hypothesize that the chosen RNA panel works in synergy to modulate STING and NOD signaling with a crucial role in T2DM pathogenesis. Also, there was a significant positive correlation between the 6 mRNAs and the important clinicopathological factors, while with the HOMA-B, there was significant negative correlation, as shown in Table 5. Also, we found a significant direct association between HOMA-IR with the chosen gene expression levels, ensuring their role in the presence of inflammation in T2DM (Table 5).

## 4. Discussion

T2DM is a very common form of DM characterized by high glucose level accompanied by insulin resistance and impairment in insulin secretion. T2DM represents around 90% of all diabetic cases [36]. Prediabetes shows high probability to developing T2DM. The different serious complications associated with chronic hyperglycemia as neuropathy, coronary heart diseases, retinopathy and nephropathy could already be seen among prediabetic patients [37]. That is why there is a great need for new methods for early detection of prediabetes.

Both the STING and NLRs pathways are mediated through different adaptor proteins, commonly found to activate the NF-kB, which induces the expression of proinflammatory cytokines. It has been suggested that STING and NLRs have a significant role in the pathogenesis of inflammation-mediated insulin resistance, which further develops metabolic complications [38]. In light of these finding, we have selected a set of six mRNAs related to STING and NLRs pathways and T2DM pathogenesis from public microarray databases. Afterwards, we studied their expression profile in sera samples to investigate its role in prediabetes and T2DM. We identified the level of expression of ZBP1, HSPA1B, TMEM173, DDX58, NFKB1 and CHUK mRNAs among all the studied groups and they are highly detected in the blood of pre-DM and T2DM patients. Raising the susceptibility of using this network as a circulating biomarker for early detection and stratification of T2DM modes may act as potential therapeutic targets.

The cGAS-cGAMP-STING pathway plays an important role in mediating inflammatory responses and may have a significant role in insulin resistance in T2DM. Qiao, J. et al. 2022 found that TMEM173 (STING1) deficiency significantly improved glucose intolerance, while neither the basal insulin level nor glucose-induced insulin secretion was increased in STING^−/−^ mice. They stated also that high-fat-diet (HFD)-fed mice (STING^−/−^ mice) showed lower levels of triacyl glycerol and cholesterol in comparison with the control group [8]. These results went hand in hand with our study that TMEM173 (STING1) was highly significant in patients with T2DM in comparison with the group with pre-DM and healthy controls. A previous study by Jiao et al., in 2020, suggested that activation of ZBP1 may lead to necrosis and chronic inflammation through triggering receptor-interacting serine/threonine-protein kinase 3 (RIPK3) [39]. A study made by Lei, Y. et al., in 2022, stated that ZBP1 can be considered a new regulator of IFN-1-mediated disease progression, acting on the cGAS-STING pathway to sense mitochondrial DNA (mtDNA) instability and sustain IFN-1 signaling, which has a role in heart failure and cardiac cell remodeling [15]. Previous studies found that single-point mutations in genes responsible for innate immunity as DDX58 might have a role in the risk of developing T1DM [40]. Consistent with our results, An.,T. et al., in 2020, performed a whole exons sequencing for obese patients with T2DM, and the DDX58 gene was one of the mutated genes that was differentially expressed in the T2DM patients [41]. On the other hand, another study found that DDX58 (RIG1) deficiency promotes insulin resistance induced by a high-fat diet in mice [42].

NLR signaling acts as critical regulator of inflammatory response triggered by the STING pathway [43]. NLRs are pathogen recognition receptors, which play a crucial role in the innate immune system that could activate the NF-kB, stimulating the expression of proinflammatory cytokines highlighting their crosstalk inflammation-mediated insulin resistance [44]. Mir et al. stated that HSP 70 gene polymorphism, the HSPA1B genotype, has been related to the severity of diabetic foot ulcers and the outcome of surgical treatment [45]. Polymorphisms in the *HSPA1B* and *HSPA1L* genes are associated with higher circulating concentrations of the inflammatory cytokines TNF-α and interleukin (IL) 6 [46]. The CHUK gene is an inhibitor of nuclear factor kappa-B kinase subunit alpha, which plays a significant role in the regulation of immune response, epidermal differentiation and keratinocyte migration [47]. Suppression of IκB kinase (IKK) and IKK-related kinases has been assessed as a potential therapeutic option for inflammatory diseases and tumors [48].

NF-κB is the core terminal effector in the pathogenesis of inflammation-mediated insulin resistance and muscle loss in diabetic patients [49,50]. Stimuli such as hyperglycemia, FFAs and reactive oxygen species could activate IκB kinase (IKK), leading to IκB ubiquitination and proteasomal degradation that can inhibit NF-κB [51]. Increased IKK/NF-κB signaling may inhibit insulin signaling through the insulin receptor, through NF-κB-mediated inflammatory proteins expression [52].

In the current pilot study, the expressions of ZBP1, DDX58, NFKB1 and CHUK were significantly higher in the T2DM group compared to either healthy control or pre-DM patients, which may give us new predictive markers for the development of T2DM in prediabetes patients. The expression of ZBP1 and NFKB1 mRNA could discriminate between the good and the poor glycemic control groups. HSPA1B mRNA showed a significant difference in its expression regarding the insulin resistance. Linear regression analysis revealed that LDLc, HSPA1B and NFKB1 were significant variables for prediction of pre-DM. To the best of our knowledge, it is a new finding that addresses the role of ZBP1, HSPA1B in the progression of T2DM. The study was limited by a relatively small sample size from a single center in Egypt. More in vitro and in vivo functional studies are needed to verify the mechanisms of RNA–RNA crosstalk in T2DM different modes.

## 5. Conclusions

In the current pilot study, we have estimated the expression of different mRNAs implicated in the cGAS-STING pathway as ZBP1, DDX58 and TMEM173 (STING1) and mRNAs involved with a NOD-like receptor pathway as HSPA1B and CHUK; all the investigated mRNA eventually affect NFKB expression, which was also measured. The mRNA panel was highly expressed in the sera of patients with T2DM in comparison with the pre-DM patients (Figure 4). These findings may have a promising impact in predicting the risk of T2DM individuals.

## Figures and Tables

**Figure 1 biomolecules-12-01230-f001:**
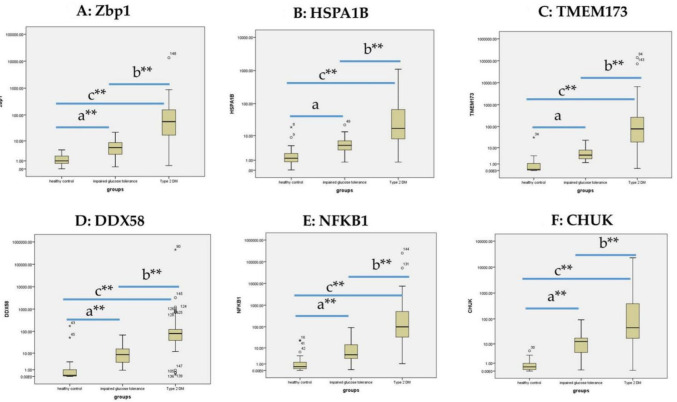
Box plot representing the relative expression of ZBP1, HSPA1B, TMEM173, DDX58, NFKB1 and CHUK mRNAs among all the studied groups. (**A**–**F**): Boxplot showing the differential expression of the 6 mRNAs among the 3 studied groups and Mann–Whitney test to analyze their expression between each 2 groups. a: The expression of the 6 mRNAs was highly significantly different between the group with pre-DM and the healthy controls. b: High significant difference in the expression of the 6 mRNAs between the T2DM group and the pre-DM group. c: High significant difference in the expression of the 6 mRNAs between the T2DM group and the healthy controls. ** highly significant *p* < 0.01.

**Figure 2 biomolecules-12-01230-f002:**
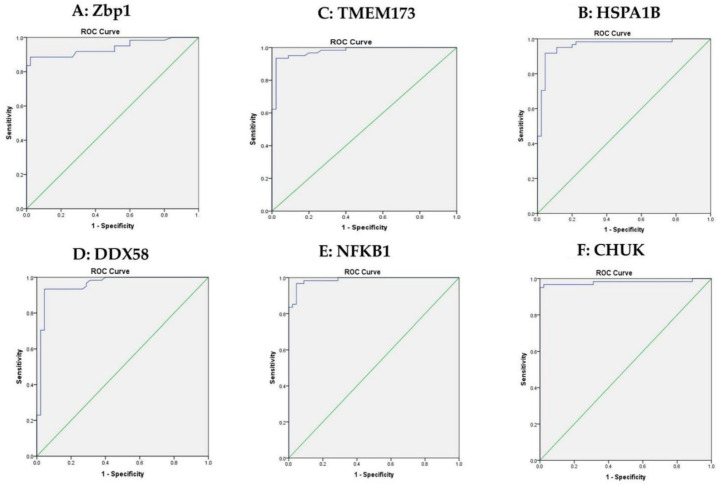
ROC curve analysis for the 6 mRNAs to discriminate between healthy controls and T2DM. (**A**): ROC curve analysis for serum Zbp1 used to calculate the best cutoff point to discriminate between T2DM and healthy controls. The best cutoff point for Zbp1 was ≥3.25 (sensitivity = 88.5%, specificity = 97.8%). (**B**): ROC curve analysis for serum HSPA1B used to calculate the best cutoff point to discriminate between the T2DM and healthy controls. The best cutoff point for HSPA1B was ≥4.7817 (sensitivity = 91.8%, specificity = 95.6%). (**C**): ROC curve analysis for serum TMEM173 used to calculate the best cutoff point to discriminate between the T2DM and healthy controls. The best cutoff point for TMEM173 was ≥4.005 (sensitivity = 93.4%, specificity = 97.8%). (**D**): ROC curve analysis for serum DDX58 used to calculate the best cutoff point to discriminate between the T2DM and healthy controls. The best cutoff point for DDX58 was ≥7.750 (sensitivity = 93.4%, specificity = 95.6%). (**E**): ROC curve analysis for serum NFKB1 used to calculate the best cutoff point to discriminate between the T2DM and healthy controls. The best cutoff point for NFKB1 was ≥6.525 (sensitivity = 96.7%, specificity = 95.6%) (**F**): ROC curve analysis for serum CHUK used to calculate the best cutoff point to discriminate between the T2DM and healthy controls. The best cutoff point for CHUK was ≥3.640 (sensitivity = 96.7%, specificity = 97.8%). Green line represents diagonal random classifier.

**Figure 3 biomolecules-12-01230-f003:**
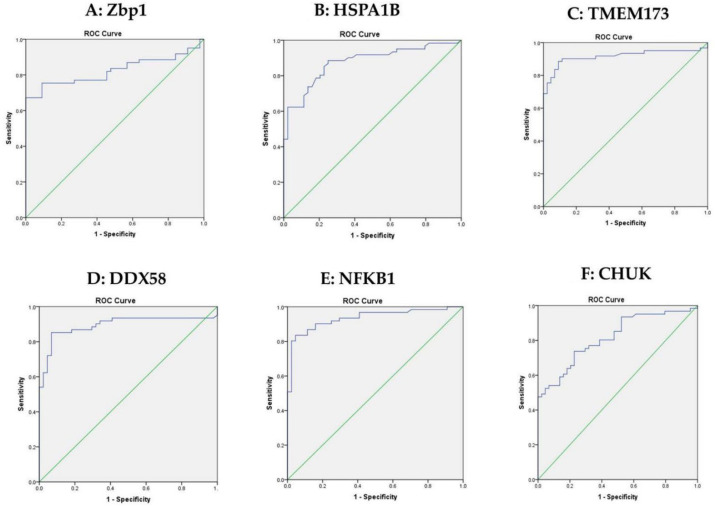
Roc curve analysis for the 6 mRNAs to discriminate between the pre-DM group and T2DM. (**A**): ROC curve analysis for serum Zbp1 used to calculate the best cutoff point to discriminate between pre-DM and T2DM. The best cutoff point for Zbp1 was ≥15.590 (sensitivity = 75.4%, specificity = 90.9%). (**B**): ROC curve analysis for serum HSPA1B used to calculate the best cutoff point to discriminate between the pre-DM group and T2DM. The best cutoff point for HSPA1B was ≥6.898 (sensitivity = 85.2%, specificity = 77.3%). (**C**): ROC curve analysis for serum TMEM173 used to calculate the best cutoff point to discriminate between the pre-DM group and T2DM. The best cutoff point for TMEM173 was ≥12.751 (sensitivity = 90.2%, specificity = 88.6%. (**D**): ROC curve analysis for serum DDX58 used to calculate the best cutoff point to discriminate between the pre-DM group and T2DM. The best cutoff point for DDX58 was ≥26.5700 (sensitivity = 85.2%, specificity = 93.2%). (**E**): ROC curve analysis for serum NFKB1 used to calculate the best cutoff point to discriminate between the pre-DM group and T2DM. The best cutoff point for NFKB1 was ≥17.350 (sensitivity = 90.2%, specificity = 84.1%) (**F**): ROC curve analysis for serum CHUK used to calculate the best cutoff point to discriminate between the pre-DM group and T2DM. The best cutoff point for CHUK was ≥17.740 (sensitivity = 73.8%, specificity = 77.3%). Green line represents diagonal random classifier.

**Figure 4 biomolecules-12-01230-f004:**
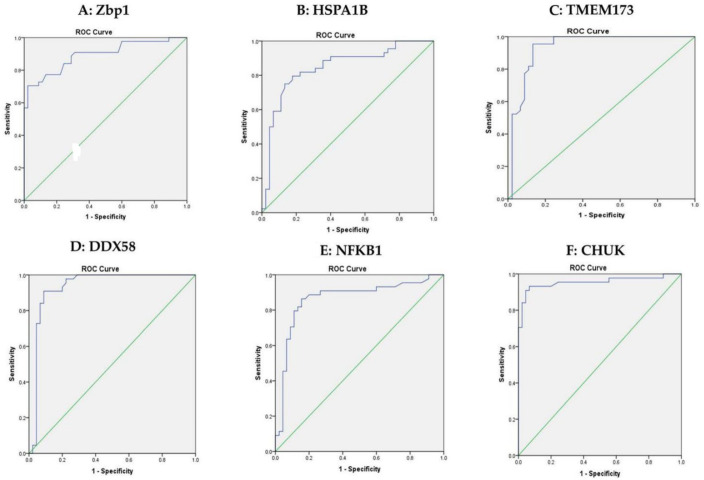
ROC curve analysis for the 6 mRNAs to discriminate healthy controls and the pre-DM group. (**A**): ROC curve analysis for serum Zbp1 used to calculate the best cutoff point to discriminate healthy controls from pre-DM. The best cutoff point for Zbp1 was ≥1.600 (sensitivity = 88.6%, specificity = 71.7%). (**B**): ROC curve analysis for serum HSPA1B used to calculate the best cutoff point to discriminate between healthy controls and the pre-DM group. The best cutoff point for HSPA1B was ≥2.341 (sensitivity = 81.8%, specificity = 77.8%). (**C**): ROC curve analysis for serum TMEM173 used to calculate the best cutoff point to discriminate between healthy controls and the pre-DM group. The best cutoff point for TMEM173 was ≥1.800 (sensitivity = 95.5%, specificity = 86.7%). (**D**): ROC curve analysis for serum DDX58 used to calculate the best cutoff point to discriminate between healthy controls and the pre-DM group. The best cutoff point for DDX58 was ≥1.220 (sensitivity = 97.7%, specificity = 77.8%). (**E**): ROC curve analysis for serum NFKB1 used to calculate the best cutoff point to discriminate between healthy controls and the pre-DM group. The best cutoff point for NFKB1 was ≥1.580 (sensitivity = 88.6%, specificity = 80%). (**F**): ROC curve analysis for serum CHUK used to calculate the best cutoff point to discriminate between the healthy controls and pre-DM. The best cutoff point for CHUK was ≥2.050 (sensitivity = 93.2%, specificity = 93.3%).

**Table 1 biomolecules-12-01230-t001:** The relative expression of the ZBP1, HSPA1B, TMEM173, DDX58, NFKB1 and CHUK mRNAs among all the studied groups.

mRNAs	Healthy Controls		Impaired Glucose Level		T2DM		χ2(c)	*p*
	Median	Mean Rank	Median	Mean Rank	Median	Mean Rank		
RQ ZBP1	1	31.37	5.3	73.37	56	109.5	83.9	0.000 **
RQ HSPA1B	1.3	32.3	4.6	68.0	17.1	112.8	90.9	0.000 **
RQ TMEM173	0.1	27.3	4.0	69.6	78.0	115.3	107.5	0.000 **
RQ DDX58	0.1	28.4	8.4	71.0	80.0	113.5	99.9	0.000 **
RQ NFKB1	0.5	29.7	4.3	65.4	99.0	116.6	106.9	0.000 **
RQ CHUK	0.4	26.2	12.5	76.7	45.0	110.9	98.6	0.000 **

Kruskal–Wallis test, *p*: *p* value, ** *p* < 0.01: Highly Significant, RQ: Relative quantity (fold change) of gene expression.

**Table 2 biomolecules-12-01230-t002:** Relation between the expression of the 6 mRNAs and the glycemic control in the T2DM group.

	Bad Glycemic Control HbA1c ≥ 7		Good Glycemic Control HbA1c < 7		U(d)	*p*
	Median	Mean Rank	Median	Mean Rank		
Zbp1	88	33	29.5	17.4	351	0.0241 **
HSPA1B	18.2	32.59	15.03	28.18	367	0.352
TMEM173	69.01	31.45	83.5	30.20	411.5	0.793
DDX58	89	32.08	71.5	29.09	387	0.528
NFKB1	122.8	33.36	66.5	19.82	437	0.05 *
CHUK	44	30.79	110.9	31.36	421	0.904

Mann–Whitney test: *p* value, ** *p* < 0.01: Highly Significant, * *p* < 0.05: Significant, *p* > 0.05: non-Significant.

**Table 3 biomolecules-12-01230-t003:** Relation between the expression of the 6 mRNAs and the insulin resistance in the T2DM group.

	Insulin Resistance HOMA-IR ≥ 2.5		Insulin Sensitive HOMA-IR < 2.5		U(d)	*p*
	Median	Mean Rank	Median	Mean Rank		
**Zbp1**	56	31.5	55.1	27.7	185.5	0.57
**HSPA1B**	88.2	32.5	12.4	17.3	134.5	0.05 *
**TMEM173**	78	30.5	105.5	34.3	185.5	0.57
**DDX58**	66.6	30.2	94	36.1	171	0.38
**NFKB1**	89	30.5	174.8	34.2	186.5	0.58
**CHUK**	45	30.2	138	36.1	171.5	0.39

Mann–Whitney test: *p* value, * *p* < 0.05: Significant, *p* > 0.05: non-Significant.

**Table 4 biomolecules-12-01230-t004:** Correlation analysis between the 6 mRNAs among all the studied groups and also among the group with pre-DM and the T2DM group.

Group			RQ (*Zbp1*)	RQ (*HSPA1B*)	RQ (*TMEM173*)	RQ (DDX58)	RQ (NFKB1)	RQ (CHUK)
**All groups**	RQ *(Zbp1)*	Correlation Coefficient	1	0.578 **	0.606 **	0.603 **	0.642 **	0.621 **
		Sig.	--------	0.000	0.000	0.000	0.000	0.000
	RQ (*HSPA1B)*	Correlation Coefficient	0.578 **	1	0.626 **	0.654 **	0.616 **	0.601 **
		Sig.	0.000	--------	0.000	0.000	0.000	0.000
	RQ (*TMEM173*)	Correlation Coefficient	0.606 **	0.626 **	1	0.740 **	0.730 **	0.739 **
		Sig.	0.000	0.000	--------	0.000	0.000	0.000
	RQ (DDX58)	Correlation Coefficient	0.603 **	0.654 **	0.740 **	1	0.691 **	0.652 **
		Sig.	0.000	0.000	0.000	--------	0.000	0.000
	RQ (NFKB1)	Correlation Coefficient	0.642 **	0.616 **	0.730 **	0.691 **	1	0.751 **
		Sig.	0.000	0.000	0.000	0.000	--------	0.000
	RQ (CHUK)	Correlation Coefficient	0.621 **	0.601 **	0.739 **	0.652 **	0.751 **	1
		Sig.	0.000	0.000	0.000	0.000	0.000	--------
**Pre-DM group and T2DM**	RQ *(Zbp1)*	Correlation Coefficient	1	0.318 **	0.374 **	0.384 **	0.463 **	0.360 **
		Sig.	--------	0.001	0.000	0.000	0.000	0.000
	RQ (*HSPA1B)*	Correlation Coefficient	0.318 **	1	0.368 **	0.475 **	0.403 **	0.351 **
		Sig.	0.001	--------	0.000	0.000	0.000	0.000
	RQ (*TMEM173*)	Correlation Coefficient	0.374 **	0.368 **	1	0.433 **	0.539 **	0.466 **
		Sig.	0.000	0.000	--------	0.000	0.000	0.000
	RQ (DDX58)	Correlation Coefficient	0.384 **	0.475 **	0.433 **	1	0.499 **	0.297 **
		Sig.	0.000	0.000	0.000	--------	0.000	0.002
	RQ (NFKB1)	Correlation Coefficient	0.463 **	0.403 **	0.539 **	0.499 **	1	0.542 **
		Sig.	0.000	0.000	0.000	0.000	--------	0.000
	RQ (CHUK)	Correlation Coefficient	0.360 **	0.351 **	0.466 **	0.297 **	0.542 **	1
		Sig.	0.000	0.000	0.000	0.002	0.000	--------

Spearman correlation. *p* value, ** *p* < 0.01: Highly Significant, *p* > 0.05: non-Significant, RQ: Relative quantity (fold change) in gene expression.

**Table 5 biomolecules-12-01230-t005:** Correlation between the 6 mRNAs and the different clinicopathological factors.

	RQ (Zbp1)	RQ (*HSPA1B)*	RQ (*TMEM173*)	RQ (DDX58)	RQ (NFKB1)	RQ (CHUK)	FSG	HbA1c	HOMA_IR	HOMA-B	BMI	Total_Cholesterol
RQ (Zbp1)												
RQ (*HSPA1B)*	0.578 **											
RQ (*TMEM173*)	0.606 **	0.626 **										
RQ (DDX58)	0.603 **	0.654 **	0.740 **									
RQ (NFKB1)	0.642 **	0.616 **	0.730 **	0.691 **								
RQ (CHUK)	0.621 **	0.601 **	0.739 **	0.652 **	0.751 **							
FSG	0.571 **	0.604 **	0.657 **	0.638 **	0.681 **	0.615 **						
HbA1c	0.505 **	0.588 **	0.567 **	0.533 **	0.652 **	0.484 **	0.671 **					
HOMA_IR	0.555 **	0.572 **	0.477 **	0.489 **	0.550 **	0.438 **	0.576 **	0.572 **				
HOMA-B	−0.655 **	−0.676 **	−0.701 **	−0.724 **	−0.734 **	−0.666 **	−0.739 **	−0.704 **	−708 **			
BMI	0.500 **	0.546 **	0.578 **	0.551 **	0.512 **	0.543 **	0.491 **	0.420 **	0.428 **	−0.495 **		
Total_Cholesterol	0.569 **	0.606 **	0.633 **	0.590 **	0.658 **	0.612 **	0.607 **	0.656 **	0.673 **	−0.733 **	0.747 **	
LDLc	0.575 **	0.669 **	0.655 **	0.640 **	0.702 **	0.622 *	0.670 **	0.655 **	0.678 **	−0.744 **	0.710 **	0.879 **

* *p* < 0.05, ** *p* < 0.01.

## Data Availability

Data presented are available on request by the corresponding authors.

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
