# Peer review of "Identification of a Multi-Messenger RNA Signature as Type 2 Diabetes Mellitus Candidate Genes Involved in Crosstalk between Inflammation and Insulin Resistance"

_biomolecules, 2022, doi:10.3390/biom12091230_

Round 1
Reviewer 1 Report
Accept for publication after minor revision
The paper aimed at evaluating circulating biomarker for early diagnosis of ‘pre-diabetes’, potentially involved in the progression from pre-diabetes to type 2 diabetes mellitus (T2DM). Authors found a set of 6 mRNAs which seems to predict pre-diabetes and T2DM.
I have the following comments and questions:
1. In the introduction (lines 47 – 55) the criteria to diagnose and classify diabetes are quite confusing, I suggest to report these, as criteria stated in ADA guidelines, both for diabetes diagnosis (fasting plasma glucose, A1c and/or 2h-post-OGTT glucose) and for differential diagnosis between type 1 and type 2 diabetes. I do not fully agree in the statement claiming it is difficult to distinguish the two forms, since both clinical presentation and demonstration of autoimmunity can help clinicians in differential diagnosis. Furthermore, since the paper investigates T2DM, talking about type 1 diabetes, beta-cell failure and autoimmunity might be misleading.
2. Since the aim of the study is to identify predictive biomarker of type 2 diabetes, in the introduction I would focus on the difficulty of an early diagnosis of T2DM and the risks associated with pre-diabetes (many patients are already affected by several chronic complications at diagnosis).
3. I suggest to better explain the STING- and NOD- signaling pathways and their involvement in inflammation and insulin-resistance (again, talking about autoimmunity might be misleading in this context).
4. In the results section, I suggest to stress the progressive and significant increase in the levels of investigated mRNAs, from healthy to pre-diabetic to diabetic patients. I also suggest to better discuss the differential expression of mRNAs in diabetic patients, according to glycemic control.
5. In the results section, I suggest to make more detailed comments to the results and significance of the correlation analysis.
6. In the discussion paragraph, lines 353 – 362, I suggest to focus on the potential benefit of an early diagnosis of prediabetes, in order to prevent diabetes-related complications, the main reason justifying the need for more accurate biomarkers of pre-diabetes.
7. English language and style editing is required. Some typing errors need to be amended.
8. Please, add a list of abbreviation used in the text.
Author Response
Reviewer 1
Accept for publication after minor revision
The paper aimed at evaluating circulating biomarker for early diagnosis of ‘pre-diabetes’, potentially involved in the progression from pre-diabetes to type 2 diabetes mellitus (T2DM). Authors found a set of 6 mRNAs which seems to predict pre-diabetes and T2DM.
- Thanks very much for your inspiring comments
I have the following comments and questions:
- In the introduction (lines 47 – 55) the criteria to diagnose and classify diabetes are quite confusing, I suggest to report these, as criteria stated in ADA guidelines, both for diabetes diagnosis (fasting plasma glucose, A1c and/or 2h-post-OGTT glucose) and for differential diagnosis between type 1 and type 2 diabetes. I do not fully agree in the statement claiming it is difficult to distinguish the two forms, since both clinical presentation and demonstration of autoimmunity can help clinicians in differential diagnosis. Furthermore, since the paper investigates T2DM, talking about type 1 diabetes, beta-cell failure and autoimmunity might be misleading.
- In agreement with reviewer comments , (lines 47 – 55) has been edited to follow ADA guidelines as follows;
- T2DM Diagnostic criteria that was clarified by the American Diabetes Association (ADA) include the following: A fasting plasma glucose (FPG) level of 126 mg/dL or higher, o rA 2-hour plasma glucose level of 200 mg/dL or higher, or A hemoglobin A1c (HbA1c) level of 6.5 or higher.
- American Diabetes Association. Classification and Diagnosis of Diabetes: Standards of Medical Care in Diabetes-2021. Diabetes Care. 2021 Jan. 44 (Suppl 1):S15-S33.
- In agreement with reviewer comments , statement claiming it is difficult to distinguish the two DM forms has been edited to be as follows;
- Although there are some difficulties in distinguishing type 1 and type 2 DM m in all age-groups at onset, but the actual diagnosis becomes more obvious over time using autoimmunity specific tests
- American Diabetes Association. Diagnosis and classification of diabetes mellitus. Diabetes Care 2014;37(Suppl. 1):S81–S90.
- In agreement with reviewer comments , sentences related to beta-cell failure and autoimmunity has been deleted
- Since the aim of the study is to identify predictive biomarker of type 2 diabetes, in the introduction I would focus on the difficulty of an early diagnosis of T2DM and the risks associated with pre-diabetes (many patients are already affected by several chronic complications at diagnosis).
- As suggested by the reviewer, a paragraph on the difficulty of early detection of T2DM was added.
- The problem is that most of the patients with type 2 DM don’t have specific symptoms in the early stage and once diagnosed the majority of the cases will have serious complications. Prediabetic state current laboratory methods show several limitations in early prediction of pre-diabetes and T2DM. The diagnosis of pre-DM relies on oral glucose tolerance test as gold standard but this test is time consuming, and complicated. Although fasting blood glucose is a convenient tool for T2DM diagnosis, the rate of missing pre-DM diagnosis is relatively high. In addition, HbA1c% is likely to be linked to other changes in red blood cell life rather than glycation rates e.g. That is why there is an urgent need to find future potential biomarkers for pre-DM early detection
- I suggest to better explain the STING- and NOD- signaling pathways and their involvement in inflammation and insulin-resistance (again, talking about autoimmunity might be misleading in this context).
As suggested by the reviewer, STING- and NOD- signaling pathways and their involvement in inflammation and insulin-resistance has been more clarified in the introduction as follows;
- cGAS–STING–IRF3 pathway is play a role in metabolic stress–induced endothelial inflammation in obesity[i].
- STING ia a critical regulator for both glucose and lipid metabolism . STING knockout Significantly improved insulin resistance and glucose intolerance in rat on high-fat diet[ii].
- NOD1 ligands lipid-derived metabolites produced during obesity and contribute to insulin resistance development[iii]. There is strong crosstalk between NOD signaling and insulin receptor pathway through NF-κB and MAPK intermediates[iv].
- In the results section, I suggest to stress the progressive and significant increase in the levels of investigated mRNAs, from healthy to pre-diabetic to diabetic patients. I also suggest to better discuss the differential expression of mRNAs in diabetic patients, according to glycemic control.
As suggested by the reviewer, the required data were added to the results as follows;
- There was a significant progressive increase in the expression levels of (ZBP1, HSPA1B, TMEM173, DDX58, NFKB1 and CHUK) from healthy to pre-diabetic by (5, 3.5, 40, 84, 14.3 and 31 folds, respectively ) and from pre DM T2DM individuals by (10.5, 3.7, 19.5,9.5, 23 and 3.6 folds respectively) .
- There was significant upregulation in the expression of (Zbp1 and DDX58 ) in poor glycemic control group compared to individuals with good glycemic control(p =0.0241 and 0.05 4, respectively) . Interestingly, HSPA1B showed higher expression in patients with HOMA-IR ≥ 2.5 compared to the other group (p=0.05).
- In the results section, I suggest to make more detailed comments to the results and significance of the correlation analysis.
As suggested by the reviewer, detailed comments to the results and significance of the correlation analysis has been added.
- There was high significant positive correlation between the 6 chosen mRNA among all studied groups and also among the T2DM group and iPre-DM group as seen in table 4. Thus, we can hypothesize that the chosen RNA panel work in synergy to modulate STING and NOD signaling with crucial role in T2DM Also, there was a significant positive correlation between the 6 mRNAs and the important clinicopathological factors, while with the HOMA-B there was significant negative correlation as shown in table 5. Also, we found a significant direct association between HOMA-IR with the chosen gene expression levels ensuring their role in in the presence of inflammation in T2DM.
- In the discussion paragraph, lines 353 – 362, I suggest to focus on the potential benefit of an early diagnosis of prediabetes, in order to prevent diabetes-related complications, the main reason justifying the need for more accurate biomarkers of pre-diabetes.
As suggested by the reviewer, a paragraph on the problems related to prediabetes and the important of early detection to prevent complications was added.
- English language and style editing is required. Some typing errors need to be amended.
As suggested by the reviewer, The whole MS has been revised for English , grammar and typing errors were corrected.
- Please, add a list of abbreviation used in the text.
As suggested by the reviewer, A list of abbreviation was added upon reviewer request.
- List of abbreviation
T2DM: Type 2 Diabetes Mellitus
STING: DNA-sensing stimulator of interferon genes
NLR: NOD like receptor
MENA: Middle East and North Africa
mRNA: messenger ribonucleic acid
NF-κB: nuclear factor-kappa beta
cGAS: cyclic GMP AMP synthase
NOD: nucleotide-binding oligomerization domain-containing protein
TMEM173: transmembrane protein 173
ZBP1: Z-DNA-binding protein 1
IRF: interferon regulatory factor
RIG-I: retinoic acid inducible gene 1
HSP: Heat shock protein
PGN: peptidoglycans
DDX58: DexD/H-Box Helicase 58
MAVs: mitochondrial antiviral signaling
MHC: Major histocompatibility complex
CHUK: conserved helix-loop-helix ubiquitous kinase
IKK α: Inhibitor-κB kinase α
mTORC1: mammalian target of rapamycin complex 1
NLRC3: NLR family domain-containing protein 3
FFA: free fatty acids
LPS: lipopolysachrides
KEGG: Kyoto Encyclopedia of Genes and Genomes
PBMCs: Peripheral Blood Mononuclear Cells
PAMPs: Pathogen associated molecular pattern
DAMPs: Damage associated molecular pattern
HSPA1B: Heat Shock Protein Family A (Hsp70) Member 1B
PPI: protein-protein interaction
ADA: The American Diabetes Association
BMI: Body Mass Index
HOMA-IR: Homeostatic Model Assessment of Insulin Resistance
FSI: fasting serum Insulin
FSG: fasting serum glucose
HOMA-B: Homeostatic Model Assessment of beta cell function
TC: total cholesterol
TG: triglyceride
HDL-c: high density lipoprotein- cholesterol
LDL-c: low density lipoprotein- cholesterol
GAPDH: glyceraldehyde 3 phosphate dehydrogenase
PCR: polymerase chain reaction
CT: cycle threshold
SPSS: The Statistical Package for the Social Sciences
ANOVA: Analysis of Variance
ROC curve: Receiver operating characteristic
HFD: high fat diet
RIPK3: Receptor-interacting serine/threonine-protein kinase 3
mtDNA: mitochondrial DNA
TNF-α: Tumor necrosis factor alpha
IL: interleukin
IKK: IκB kinase
[i] Mao Y, Luo W, Zhang L, Wu W, Yuan L, Xu H, Song J, Fujiwara K, Abe JI, LeMaire SA, Wang XL, Shen YH. STING-IRF3 Triggers Endothelial Inflammation in Response to Free Fatty Acid-Induced Mitochondrial Damage in Diet-Induced Obesity. Arterioscler Thromb Vasc Biol. 2017 May;37(5):920-929. doi: 10.1161/ATVBAHA.117.309017. Epub 2017 Mar 16. Erratum in: Arterioscler Thromb Vasc Biol. 2018 Apr;38(4):e60.
[ii] Qiao J, Zhang Z, Ji S, Liu T, Zhang X, Huang Y, Feng W, Wang K, Wang J, Wang S, Meng ZX, Liu M. A distinct role of STING in regulating glucose homeostasis through insulin sensitivity and insulin secretion. Proc Natl Acad Sci U S A. 2022 Feb 15;119(7):e2101848119.
[iii] Rivers SL, Klip A, Giacca A. NOD1: An Interface Between Innate Immunity and Insulin Resistance. Endocrinology. 2019 May 1;160(5):1021-1030. doi: 10.1210/en.2018-01061. PMID: 30807635; PMCID: PMC6477778.
[iv] Chan KL, Tam TH, Boroumand P, Prescott D, Costford SR, Escalante NK, Fine N, Tu Y, Robertson SJ, Prabaharan D, Liu Z, Bilan PJ, Salter MW, Glogauer M, Girardin SE, Philpott DJ, Klip A. Circulating NOD1 activators and hematopoietic NOD1 contribute to metabolic inflammation and insulin resistance. Cell Reports. 2017;18(10):2415–2426

Reviewer 2 Report
The research presented by the authors is novel and with probable clinical application. Their results are interesting; however, there are some points to consider before publication.
1. The first paragraph of the methodology section seems more part of the background.
2. In general, the statistical analysis is not correct. For example, they mention in table 1 of the text and supplementary table 1 the results were analyzed with the Mann-Whitney U and Student's t-tests, but in both cases the authors present three groups to be compared. Using these tests invalidates the results because the analysis is not correct.
3. It is necessary to expand the explanation of Figure 5, in order to give a better conclusion.
4. It is necessary a great work in the edition of the tables and graphs, they are badly centered, not very legible, without justification, etc.
5. I consider that the supplementary table 2 should be changed to the text, its results are interesting, but the explanation and discussion of this table should be expanded.
6. Throughout the document there are multiple typos, different handwriting, and other spaces. This also applies to the references section.
Author Response
Reviewer 2
The research presented by the authors is novel and with probable clinical application. Their results are interesting; however, there are some points to consider before publication.
- Thanks very much for your valuable comments
- The first paragraph of the methodology section seems more part of the background.
As suggested by the reviewer, the first paragraph of the methodology has been moved to introduction
- In general, the statistical analysis is not correct. For example, they mention in table 1 of the text and supplementary table 1 the results were analyzed with the Mann-Whitney U and Student's t-tests, but in both cases the authors present three groups to be compared. Using these tests invalidates the results because the analysis is not correct.
- As suggested by the reviewer, We have deleted Mann-Whitney U from table 1 footnote
- We have used Kruskal-Wallis test to explore the statistical signicance among the three groups. Meanwhile, we also used Mann-Whitney U to explore the differences between each 2 groups inside the 3 studied groups(Pre-DM group and the healthy controls, T2DM group and the pre-DM group , T2DM group and the healthy controls) that has been found in figure 1.
- It is necessary to expand the explanation of Figure 5, in order to give a better conclusion.
- As suggested by the reviewer, explanation of Figure 5 has been added as follows;
- Chronic inflammation triggers stimulation of the expression of cGAS-STING pathway crucial players as ZBP1, DDX58 and TMEM173 (STING1) mRNAs and also overexpression of NOD like receptor pathway effectors as HSPA1B and CHUK mRNAs. Activation of both pathways leads to stimulation of NFKB expression that leads to insulin resistance and T2DM pathogenesis
- It is necessary a great work in the edition of the tables and graphs, they are badly centered, not very legible, without justification, etc.
As suggested by the reviewer, we have tried to edit graphs and tables
- I consider that the supplementary table 2 should be changed to the text, its results are interesting,
- As suggested by the reviewer, supplementary table 2 has been clarified in text in the results section as follows
Linear regression analysis revealed that the most significant variable for prediction of preDM was LDLc (p value 0.001), (Standardized Coefficients Beta 0.429 followed by HSPA1B ( p value 0.014) ,( Standardized Coefficients Beta 0.118) ; then NFKB1 ( p value 0.031) ,( Standardized Coefficients Beta 0.113).
- Throughout the document there are multiple typos, different handwriting, and other spaces. This also applies to the references section.
As suggested by the reviewer, The whole MS has been revised for English , grammar and typing errors were corrected.

Round 2
Reviewer 2 Report
The authors improved the manuscript and responded to my comments.
However, Figure 1 is still not legible, especially the legend of the groups, the text is too small.